# Management of Porcelain Gallbladder, Its Risk Factors, and Complications: A Review

**DOI:** 10.3390/diagnostics11061073

**Published:** 2021-06-10

**Authors:** Masaya Morimoto, Takahiro Matsuo, Nobuyoshi Mori

**Affiliations:** Department of Infectious Diseases, St. Luke’s International Hospital, Tokyo 104-8560, Japan; tmatsuo@luke.ac.jp (T.M.); morinob@luke.ac.jp (N.M.)

**Keywords:** porcelain gallbladder, cholelithiasis, gallstone, gallbladder cancer, cholecystectomy

## Abstract

The porcelain gallbladder condition describes gallbladder calcification. While gallbladder calcification is believed to increase the risk of developing gallbladder cancer, recent reports have shown that the malignancy risk is much lower than previously reported. Symptomatic patients with porcelain gallbladder should be recommended for cholecystectomy, but the management of asymptomatic patients is debatable. Based on recent evidence, prophylactic cholecystectomy is not routinely recommended in all patients with porcelain gallbladder. From the assessment of the current literature, there are three essential factors in the management of patients with porcelain gallbladder: (1) symptoms or complications of gallbladder disease, (2) calcification pattern and (3) patient age and comorbidities. Patients who do not undergo cholecystectomy should be educated about the symptoms of gallbladder diseases, and a thorough discussion is essential between patients and clinicians.

## 1. Introduction

Gallbladder calcification is known as porcelain gallbladder (PGB). The term PGB originally refers to the blue discoloration and brittle consistency of the gallbladder wall, but it is often used to describe all types of gallbladder calcification. When extensive calcium deposits invade the gallbladder, the gallbladder wall can become fragile, brittle, and bluish, which results in a porcelain appearance. Other names for this condition are calcified gallbladder or calcifying cholecystitis. PGB is classified into two types based on the extent of calcification: complete intramural calcification or selective mucosal calcification. The latter type has been reported to be associated with the development of gallbladder cancer (GBC) [1,2]. The pathogenesis of PGB is controversial, but it has been hypothesized that gallbladder calcification is caused by chronic inflammation [3]. Gallbladder calcification occurs in association with gallstones, with many PGB patients being asymptomatic, and is detected through incidental radiographic findings. The diagnosis of PGB is mainly based on abdominal radiography or ultrasonography, confirmed by surgical and pathological results. 

According to traditional methods, when PGB is diagnosed, prophylactic cholecystectomy should be performed to prevent malignancy. Older reports [4,5] suggested that PGB is a very high-risk factor with a predisposition to the development of GBC; thus, cholecystectomy has become a standard method in the management of patients with PGB. However, more asymptomatic patients are diagnosed incidentally on abdominal imaging with increased accuracy and frequency of imaging tests, and recent studies [1,6] have shown that the incidence rate of GBC in PGB is much lower than previously expected. This change has raised questions regarding routine cholecystectomy in patients with PGB. Since cholecystectomy for PGB is difficult due to the brittle and calcified gallbladder, the risk of perioperative complications and conversion to open surgery is not low [7]. For patients with a perioperative risk or mortality that is higher than the risk of developing malignancy, prophylactic cholecystectomy should not be recommended, and conservative management is advised. Clinicians are concerned about the risk of malignancy among patients with PGB, but it is still not clear, and the management of these patients is a controversial issue. PGB is a very rare disease, and it is challenging to conduct randomized controlled trials. Therefore, there is less evidence on the indications of cholecystectomy in such cases. Based on a literature review, we discussed the characteristics of PGB, its association with GBC and the best management techniques in these patients.

## 2. Epidemiology

PGB is rare, with an incidence of <1% in patients with gallbladder diseases [4,5,8]. It is approximately five times more prevalent in females [4], and most cases occur over the age of 60 years. Gallstones are present in approximately 60% to 90% of cases [1,6]; thus, cholelithiasis is a significant risk factor in the development of calcified gallbladder wall. More than 85% of gallstones are cholesterol stones in patients of the developed countries, and approximately 20 million people (14 million women and 6 million men) in the United States have gallstones [9]. Advanced age [10], female sex [11], ethnic differences (Mexican Americans) [12] and genetic factors [13] are major risk factors of cholesterol gallstones. Patients with severe weight loss or fasting have a high risk of gallstones due to biliary stasis [14,15]. Hormonal association with gallstones also exists, and estrogen is correlated with an increase in bile cholesterol and a decrease in gallbladder contractility [11,16]. This may be one of the reasons for its predilection in women. Chronic diseases, such as diabetes mellitus, also cause an increase in gallstone formation due to neuropathy following reduced gallbladder wall contractility [17,18]. The etiologies that cause gallstones and the prolonged presence of cholelithiasis are major factors in the development of PGB. Although some patients with PGB occasionally present with symptoms due to complications of gallbladder diseases, others are often asymptomatic. According to previous reports, 18–33% of patients with PGB were asymptomatic cases [1,6,19]; however, recent studies indicated that asymptomatic PGB patients accounted for 70–86% of the study population [7,20]. 

## 3. Calcification Patterns

PGB is characterized by gallbladder calcification, and histological examination shows that calcification is distributed throughout the mucosa, submucosa, glandular spaces and Rokitansky-Aschoff sinuses [1,21]. Calcification of the gallbladder is classified based on its extent: complete intramural calcification and selective mucosal calcification. The gallbladder wall undergoes complete calcification and fibrosis in the complete type; however, in the selective type, the calcification is milder and restricted to the mucosal layer. In a study of 44 patients with PGB [1], 17 showed complete intramural calcification and 27 showed selective mucosal calcification. No patients with the complete type had GBC, but two patients with the selective type had GBC. The absence of mucosa in the complete type may reduce the risk of malignancy, and a systematic review also indicates that the selective type is significantly positively associated with gallbladder malignancies [2]. Khan et al. identified the complete intramural type and selective mucosal type in 69% and 23% of 13 patients with PGB, respectively [3].

## 4. Radiological Findings

PGB cases may be detected incidentally on abdominal imaging. Plain abdominal radiographs may show rim-like calcifications in the gallbladder wall [22]. Imaging modalities for PGB include ultrasound [23] and abdominal computed tomography (CT) scans [24], and diffusion-weighted magnetic resonance imaging has been reported to be effective in differential diagnosis [25]. The radiographic findings of calcified gallbladders vary according to the location, extent and degree of calcification. Less intense calcification is difficult to identify with plain radiography, but more diffuse severe calcification appears as rounded or curvilinear findings, as reported by a case report [22]. Recently, the frequent use of ultrasonography in patients with abdominal problems has led to earlier detection of PGB. The ultrasonography findings of PGB have been classified into three types based on the extent and nature of calcifications [5]. Type I is a hyperechoic semilunar structure with posterior acoustic shadowing, type II is a curvilinear echogenic structure with acoustic shadowing, and type III are irregular clumps of echoes with posterior acoustic shadowing. Type I corresponds to complete intramural calcification, while type II and type III show the variations of selective mucosal calcification. However, in some cases, minor lesions cannot be detected on radiological tests but are found on pathology. 

The accuracy of CT scans in diagnosing PGB was evaluated by Appel et al. [24]. PGB was reported in 133 CT studies by two independent radiologists, and the diagnosis was confirmed by surgery, pathology or follow-up imaging. Pathology results confirmed PGB in 90/133 (68%) patients, but one-third (42/133; 32%) of CT reports did not confirm PGB. In this study, frequent misleading causes were stones filling the whole gallbladder lumen in 91%, sludge in 7% and mucosal enhancement in 5%. CT is the most sensitive method of detecting calcification; thus, it may cause an overdiagnosis of PGB. It is essential for clinicians to know the frequent pitfalls associated with the assessment of imaging.

## 5. Pathogenesis

The exact pathogenesis of gallbladder calcification is still unclear, but it is thought to be the final result of chronic inflammation resulting in hemorrhage, scarring and hyalinization of the gallbladder wall [3]. In other words, it is a morphological variant of chronic cholecystitis. The main etiopathology is thought to be chronic gallbladder wall irritation caused by gallstones and cystic duct obstruction with bile stagnation, which produces mucosal calcium carbonate precipitation [8,26]. Other possible mechanisms include dystrophic calcification, errors in calcium metabolism, inflammation and ischemia [8,26]. Bile stasis might be a chemical carcinogen [27]. Petrov et al. induced gallbladder carcinoma by implanting hard foreign bodies in the gallbladder wall of animals [28]. Obstruction of the cystic duct produces precipitation of calcium salts in the mucosa followed by degeneration and regeneration of chemicals in stagnant bile salts on the gallbladder epithelium. This process may lead to mucosal dysplasia and progression of cancer. Although squamous cell carcinomas have also been described, most carcinomas associated with PGB are adenocarcinomas [29]. Although PGB and GBC are highly correlated with chronic inflammation [30] and gallstones, there is no conclusive evidence of their direct association. It is difficult to demonstrate the causal relationship between gallstones, calcification, PGB and GBC [31]. Although there are still many unknown factors, some interesting findings have been reported recently. Dorvash et al. studied the correlation between metformin and gallstone disease [32]. They showed that metformin decreases the chances of developing gallstone disease in animal models, but they also observed that almost all metformin-treated mice developed mucosal calcification, which mimics PGB. Further investigation is needed to explore the relationship between metformin consumption and gallbladder mucosal calcification in mice and humans.

## 6. Differential Diagnosis

The list of differential diagnoses of right upper quadrant calcification includes a large gallstone, echinococcal cysts, calcified renal cysts, chest wall masses with calcification, degenerative cystic lesions of the pancreas, calcified adrenal tumors or an atherosclerotic aneurysm of the abdominal aorta [33]. 

In the differential diagnosis of these diseases, ultrasonographic findings of gallstones or “gas” may play an important role. First, cholelithiasis and PGB are both causes of shadowing in the gallbladder fossa. However, in cases with cholelithiasis, there is a thin hypoechoic bile space between the gallbladder wall and gallstone echo. This ultrasonographic “wall-echo-shadowing sign” (also known as WES sign) [34] suggests that a large gallstone or multiple small gallstones fill the lumen of the gallbladder. Second, patients with emphysematous cholecystitis have an echogenic crescent in the gallbladder fossa, and the ultrasound findings of shadows from gas within the gallbladder are similar to calcification in patients with PGB. However, emphysematous cholecystitis or pneumobilia produce “dirty” acoustic shadowing and “ring-down” artifact rather than the clean anechoic shadow of calcification [35].

## 7. Risk of Malignancy

Traditionally, PGB was believed to be strongly associated with GBC, and the rate of GBC occurring in PGB was in the range of 12% [4]–33% [5]. However, the actual rate was reported to be lower in recent studies (0% [6]–5% [1]). A systematic review of 111 articles containing data on 340 patients with gallbladder calcification showed that the incidence of GBC was 21% [2]. Although the incidence rate in the review is high, we have to evaluate the data carefully because some limitations cause overestimation of the incidence of malignancy. There are three common biases: (1) publication bias, in which publishers might prefer to report only rare or curious cases, (2) selection bias, due to selecting subjects from a population of GBC patients rather than the general population, and (3) sampling bias, due to the inclusion of symptomatic patients seeking medical advice for gallbladder-related symptoms. Most cases of gallbladder calcification are asymptomatic; thus, many benign cases are undiagnosed. By eliminating these factors in a subgroup analysis of 13 studies without selection bias, the rate of GBC was 6% in patients with PGB; in contrast to 1% in patients without PGB (relative risk: 8.0; 95% confidence interval (CI): 1.0–63.0) [2]. 

Many reports suggest that the association between PGB and GBC can be found in focal calcification rather than extensive calcification; however, the depth of calcification has not been reported to be related to malignancy [1,2]. It is necessary to predict the risk of malignancy because GBC is highly fatal [36], but patient age, calcification limited to a focal area of the gallbladder wall, and the absence of microcrystals or stones in the gallbladder are not found to be significant risk factors of GBC in PGB [2]. Typical symptoms of GBC (painless jaundice, Courvoisier’s sign, and unexplained weight loss) and gallbladder mass are statistically significant risk factors (odds ratio [OR]: 83.6; 95% CI: 2.3–2979 and OR: 3226.6; 95% CI: 17.2–603,884.8, respectively) [2]. Both typical GBC symptoms and gallbladder masses are signs of advanced status; therefore, these findings cannot be used for the early detection of malignancy.

## 8. Literature Review

We reviewed 567 articles from the PubMed database and 1009 articles from the EMBASE database (314 of which were not MEDLINE data) from 1950 to March 2021. The following search terms were used: (“gallbladder*” OR “gall bladder” OR “cholecystosis” OR “cholecystitis”) AND (“porcela*” OR “calcif*” OR “calcinosis” OR “calcareous” OR “ossifi*”). From these articles, we focused on review articles that added the search terms “review” [publication type] OR “systematic review” [publication type] OR “review” [text word]. We found 87 articles, but excluded those that did not mention PGB or were not written in English, and finally reviewed 12 articles [2,3,6,7,20,24,37,38,39,40,41,42]. Furthermore, we retrieved 12 articles and case series [1,3,4,5,6,7,19,20,37,43,44,45] showing the incidence of PGB with GBC (Table 1). 

Cornell et al. reported that 4271 cholecystectomies performed at a single center in New York between 1922 and 1956 revealed 16 calcified gallbladders, of which two (12.5%) had concomitant GBC [4]. In 1967, an Argentinean study demonstrated that 16 of 26 (61.5%) patients with PGB had GBC based on 1786 cholecystectomies [46]. Kane et al. reported a 33% concordance between PGB and GBC in 1984 [5]. However, recent studies have reported a decline in the incidence of GBC among patients with PGB [1,6]. Stephen et al. reported that 25,900 cholecystectomies performed between 1962 and 1999 accounted for 4.5% (2 of 44 cases) of GBC in PGB [1]. This study mentioned that the association between PGB and GBC is strongest with mucosal wall calcification but lower with transmural calcification. Another study [6] of 10,741 specimens between 1955 and 1998 showed similar results. None of the 15 cases with PGB had GBC, and when 88 cases of GBC were analyzed, none of them had PGB. 

From this literature review, we can consider that the association between PGB and GBC is less common recently than in the past. The following hypotheses might explain the decline in malignancy incidence: (1) our changes in diet or the environment may transform the natural history of the disease; (2) modern imaging modalities have been developed, and frequent imaging tests make it easier to diagnose asymptomatic cases; (3) the availability of minimally invasive surgery causes many people to undergo cholecystectomy before malignancy develops; (4) most importantly, PGB is extremely rare, and the sample size is low. For example, in a small cohort (20 patients), the reported rate of malignancy would increase from 0% to 5% or 10% if only one or two patients were diagnosed with cancer. When the number of cases is less, a single case diagnosis may inflate the association between PGB and GBC. Recent studies have made efforts to be careful and provide large patient cohorts. 

## 9. Prognosis and Management

Patients with nonmalignant PGB treated with cholecystectomy have the same good prognosis as those with cholecystitis treated with routine surgery. In any case, the prognosis of PGB depends on whether GBC complicates it. Although GBC is a relatively rare malignancy, it is considered a highly lethal disease. The incidence is 0.1–1.2%, and it is the most common biliary tract tumor and the fifth most common neoplasm of the gastrointestinal tract [47]. The prognosis of GBC is extremely poor because the patient mostly presents with symptoms at an advanced stage of the condition. The five-year survival rate is less than 5% [48,49], while the median survival is 9.2 months for patients with suspected malignancy and 26.5 months for those with incidentally diagnosed malignancy [50]. Therefore, clinicians are always concerned about the risk of malignancy in patients with PGB. If these PGB patients are symptomatic, or have gallbladder disease complications, even in the absence of malignancy, surgical management should be suggested. However, when PGB patients are asymptomatic and diagnosed incidentally, it is difficult to decide on the management protocol. Among asymptomatic PGB patients, the role of prophylactic cholecystectomy is arguable [2,43,51,52]. A study by Appel et al. showed that no GBC developed in patients with CT diagnosis of PGB during 6.6 ± 4.6 years of follow-up [24]. Further, no patients with PGB developed cancer within 3.2 ± 3.2 years of follow-up in the study by DesJardins et al. [20] and 3.5 years of follow-up in the study by Chen et al. [7]. Desjardins et al. also showed that there were no differences in outcomes between patients who underwent cholecystectomy and those who were subjected to observation [20].

Considering all reports so far, the management of PGB is controversial. The clinical practice guidelines of the European Association for the Study of the Liver [53] states that the evidence supporting prophylactic cholecystectomy in PGB patients is of low quality and remarks that it “may be avoided in patients with homogenous wall calcification,” in other words, the complete intramural type, which has been reported to not increase the risk of malignancy compared to the other type [1,2]. When considering the surgical approach for PGB, the following three factors are important: (1) symptoms or complications of gallbladder disease such as right hypochondrial pain, common duct obstruction, cholangitis or recurrent pancreatitis; (2) calcification pattern, i.e., selective mucosal type versus complete intramural type and (3) the patient’s age and comorbidities. Figure 1 depicts a flowchart on the management of PGB. 

Considering the calcification type, patients with complete type PGB should be referred for cholecystectomy when they have any symptoms or complications of gallbladder disease. Patients with a selective mucosal type should be considered for surgery even if they have no symptoms. Prophylactic cholecystectomy adds the benefit of eliminating undetected malignancies by removing the gallbladder. Thus, this results in a better outcome in these patients. In young and fit patients, cholecystectomy is a good option; however, it is not recommended for those who have a high risk of perioperative mortality, and conservative management is better in them. It is essential to carefully assess the risk of developing malignancy and the risk of perioperative complications. Patients who do not undergo cholecystectomy may require close follow-up even though there are no actual recommendation rules. A calcified gallbladder might make it difficult to eject bile, and cause gallstone formation. These patients should be educated about gallstone and gallbladder disease symptoms and seek medical services when any symptoms occur.

## 10. Surgical Treatments

While surgeons perform cholecystectomy, it is essential to subject the surgical specimen to a frozen section and perform histopathological examinations. If this reveals malignancy, conversion to an extended or radical cholecystectomy is needed (wedge resection of the liver and gallbladder bed, followed by lymphadenectomy). When prophylactic cholecystectomy is performed in patients with PGB, a laparoscopic approach is considered appropriate. However, some reports show that laparoscopic cholecystectomy is a difficult procedure in patients with PGB because the brittle and calcified gallbladder wall is difficult to grasp with laparoscopic forceps [19,54]. This surgery carries a risk of blood loss, bile duct injury, and conversion to open surgery, but recently, this surgery has been performed safely because of the development of laparoscopic instruments [3]. In addition to conventional laparoscopic cholecystectomy, single-incision laparoscopic cholecystectomy has been reported [52]. The single-incision approach is more difficult than the conventional approach, but has better cosmetic benefits [55,56]. Some reports have indicated that the conversion rate to open surgery is 5–25% [3,7,39]. Khan et al. showed that obtaining an adequate view of the cystic duct and artery was the leading cause of conversion to open surgery [3]. The major complication risk of cholecystectomy has been reported to be only 3–4%, including perioperative mortality of 0.5% and common bile duct injury of 0.5% [57,58,59,60,61]. Although the risk of surgery is relatively low in most patients, some patients with comorbidities have a higher risk of cholecystectomy. For example, cirrhosis is associated with an increased risk of poor outcomes after cholecystectomy. Laparoscopic cholecystectomy in cirrhotic patients is associated with a higher complication risk than in noncirrhotic patients, and the severity of complications is related to the Child-Pugh class, with a maximum in class C [61]. In a study that assessed the surgical outcome of PGB, of 192 patients, 102 underwent cholecystectomy and 90 were under observation [7]. Although 82% of the patients were asymptomatic in the operative group, they underwent surgery because of concerns regarding GBC. The perioperative complication rate among asymptomatic patients was 10.7%, but that among symptomatic patients was 16.7%. These outcomes in patients with PGB depend on symptoms prior to cholecystectomy, and this study, showed a much higher complication rate in PGB than that in cholecystectomy for other diseases (3–4% as mentioned above) because operating on the brittle, calcified gallbladder is technically difficult. The perioperative complications in this study included bile leak, infection, common bile duct stricture and incisional hernia, which sometimes led to additional endoscopic or percutaneous interventions and resurgeries. Based on these data, we do not recommend prophylactic cholecystectomy in all asymptomatic patients with PGB. 

## 11. Conclusions

Recent evidence has suggested that the risk of developing GBC among patients with PGB is much lower than previously believed. Prophylactic cholecystectomy should not be routinely performed in asymptomatic PGB patients but may be considered only when patients have any indications for cholecystectomy. If patients with PGB have any typical symptoms or are young and fit, the optimal management should be prophylactic cholecystectomy. Calcification pattern is also crucial for considering indications of surgery, and asymptomatic patients with selective mucosal calcification may need to consider surgical treatment because of their high potential for malignancy. In conclusion, considering the surgical approach, the following factors should be evaluated: (1) symptoms or complications of gallbladder disease such as right hypochondrial pain, common duct obstruction, cholangitis or recurrent pancreatitis; (2) calcification pattern, i.e., selective mucosal type versus complete intramural type and (3) patient’s age and comorbidities. When patients are elderly or have some comorbid conditions, a discussion is needed to compare the risk of perioperative complications compared to the low risk of malignancy. For those people, observation might be adequate, and the nonoperative approach may require close follow-up. They should be educated about PGB, GBC and gallbladder diseases to seek medical services when any symptoms occur. 

## Figures and Tables

**Figure 1 diagnostics-11-01073-f001:**
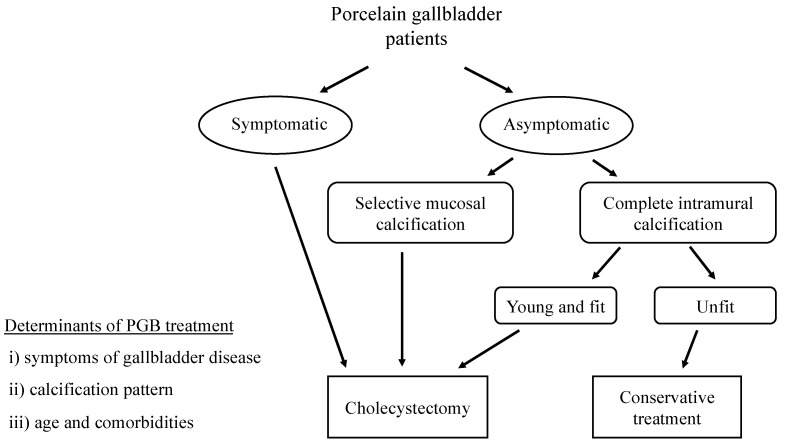
Management of porcelain gallbladder patients.

**Table 1 diagnostics-11-01073-t001:** Review of literature: cross-sectional studies evaluating the percentage of patients with porcelain gallbladder (PGB) who developed gallbladder cancer (GBC).

First Author (Year) [References]	Country	Samples	PGB CasesN (%)	GBC in PGBN (%)	Asymptomatic Cases, N (%)
Cornell et al. (1959) [4]	United States	4271	16 (0.4)	2 (12.5)	N/A
Kane et al. (1984) [5]	United States	-	9 (-)	3 (33.3)	6 (67)
Gale et al. (1985) [37]	United States	-	4 (-)	0 (0.0)	N/A
Towfigh et al. (2001) [6]	United States	10,741	15 (0.1)	0 (0.0)	5 (33)
Stephen et al. (2001) [1]	United States	25,900	44 (0.2)	2 (4.5)	8 (18)
Kwon et al. (2004) [19]	Japan	1608	13 (0.8)	1 (7.7)	4 (31)
Kim et al. (2009) [43]	Korea	3159	9 (0.3)	0 (0.0)	N/A
Khan et al. (2011) [3]	United states	1200	13 (1.1)	0 (0.0)	0 (0)
Hayes et al. (2014) [44]	Ireland	2522	5 (0.2)	0 (0.0)	N/A
Chen et al. (2015) [7]	United States	-	192 (-)	0 (0.0)	166 (86)
Kapoor et al. (2016) [45]	India	116	1 (0.9)	0 (0.0)	N/A
DesJardins et al. (2018) [20]	United States	-	113 (-)	0 (0.9)	79 (70)
Total			434	8 (1.8)

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
