# Peer review of "Management of Porcelain Gallbladder, Its Risk Factors, and Complications: A Review"

_diagnostics, 2021, doi:10.3390/diagnostics11061073_

Round 1
Reviewer 1 Report
In this review, 12 articles regarding PGB were reviewed to describe the characteristics of PBG, its association with GBC, as well as 12 articles and case series concerning the incidence of PGB with gallbladder cancer.
The authors concluded that based on recent literature, the incidence of PGB with gallbladder cancer is lower than previously expected. Also, they noted three factors essential for the management of PGB, including presence of symptoms, calcification pattern and the fitness of patients.
Overall, the manuscript is in well-written English, with extensive description of the literature and background. There are some spelling checks needed and some parts are overlapping and may be removed or replaced. Below are the specific comments mentioned:
1. Introduction section: the introduction section is extensive and relatively long; especially the first paragraph might be shortened.
- Line 42, page 1; 'incidence rate': please specify which incidence rate (PGB, or GBC in PBG?)
- Line 45-48, page 2: how can these risks be determined? Can it even be determined? Reference is missing to support this statement.
2. Epidemiology: well written, no comments
3. Idem
4. Radiological findings:
- Line 105-106, page 3: this sentence seems more appropriate in the calcification section (3).
- Line 112-113, page 3: this sentence fits better in the risk of malignancy section (7).
- Line 113-116, page 3: the calcification patterns and their definitions have already been mentioned before, so these sentences can be shortened.
5. Pathogenesis
- Line 133-137, page 3: this part might be more appropriate as part of a 'pathological findings' section, for example combined with section 3.
6. Differential diagnosis: clear and extensive description signs that might indicate PGB or other diagnosis.
7. Risk of malignancy:
- Line 173, page 4: replace 'because of' by 'due to', since it is a negative consequence.
- Line 177-179, page 4: rephrase this sentence, for example: 'By eliminating' these factors in a ... in contrast 'to' 1%.
- Line 180-183, page 4: this is also mentioned in section 8, line 226-231, so this might be removed.
8. Literature review: extensive literature review, raising some questions:
- What time frame was used when reviewing the databases? Please mention in manuscript.
- A flow chart describing the inclusion and exclusion of articles would be great to have an overview.
- Regarding line 204, page 4: references are not well-organized (i.e. 38-42 instead of 42, 43, 38-41). Moreover, some of these references, such as 42 and 43, can not be found elsewhere in manuscript. Which data or info was used from these references?
- Why did the authors chose to perform two literature searches/questions? Since the incidence has been described previously in a similar table in the review of Machado (ref 42), this question and the corresponding Table 1 seem redundant. It would be better to refer to this reference in text with the authors own additional information.
9. Prognosis and management
- Line 253, page 6: where was this percentage mentioned before in manuscript?
- Line 254-256, page 6: this information belongs to section 7 and 8 (and Table 1?)
- Line 287-288, page 7: do the authors have any recommendations for follow-up (imaging methods, time period)
10. Surgical treatments
- Line 297, page 7: I believe the authors mean lymphadenectomy?
- Do the authors have any recommendations for which centers should perform such surgery (only tertiary referral hospitals?)
- Are there any recommendations regarding laparoscopic or open cholecystectomy?
11. Conclusion: conclusions are well written, but a bit too long. Please abbreviate this section.
- Line 334-336, page 8: please rephrase this sentence; it is confusing since the authors state that no cholecystectomy should be performed in asymptomatic patients, while in the manuscript and below this sentence they do state that it should in some patients. Perhaps 'unless' is a better option.
- Line 338-340: add asymptomatic in this sentence since this is unclear now.
References: please check references since some are in capital letters.
Reviewer 2 Report
1. From the assessment of the current literature, there are three essential factors in the management of patients with porcelain gallbladder: 1) symptoms or complications of gallbladder disease, 2) calcification pattern, and 3) patient age and comorbidities.
In this sentence 3 factors are described for assessment of these patients not for management as there are no treatment option summarized.
2.
However, more asymptomatic patients are diagnosed incidentally on abdominal imaging 40 with increased accuracy and frequency of imaging tests, and recent studies [1,6] have 41 shown that the incidence rate is much lower than previously expected.
I don't understand why the incidence rate of calcified galbladder should change the decision whether or not cholecystectomy should be considered.
3. Plain abdominal radiographs may show rim-like calcifications in the gallbladder wall [21], and not the specific findings of PGB
Elaborate these specific findings? Or rephrase this sentence
4. Please change "minute" lesions into minor lesions on line 104.
5. The difference strategies are discussed to diagnose, typify porcelain galbladder. It would be nice if a table would be introduced described the advantages, disadvantages of each technique including the subgroups of porcelain galbladder/ or different patterns they can discern.
6. It would be nice to add pictures of the different types (type I-III) visual on radiological tests and patterns visual on histopathology.
7. CT is the most sensitive method of detecting calcification; thus, it may cause an overdiagnosis of PGB
What is meant with overdiagnosis? To my knowledge it seems best to make correct diagnoses as much as possible. Do you mean that the CT is not specific and sometimes make wrong diagnoses.
8. Bile stasis may be a chemical carcinogen
I am not sure you can state this. As bile stasis is the phenomenon of the general liver disease cholestasis, which could evolve into cancer, albeit there is only a minor percentage that effectively progresses to cholangiocarcinoma.
9. To eliminate these factors, in a subgroup analysis of 13 studies without selection bias, the rate of GBC was 6% in patients with PGB; in contrast, 1% in patients without PGB (relative risk: 8.0; 95% confidence interval (CI): 1.0-63.0) [2].
I don't understand how you eliminated the biases. Could you elaborate.
10. This decreasing incidence of cancer might be due to the possible changes in the diet or environment on early detection of calcification by modern investigation methods.
I thought this was because you removed the biases?
11. Both typical GBC symptoms at line 192. You described in the sentence earlier 3 symptoms..
12. None of the 15 cases with PGB had GBC, and when 88 cases of GBC were analyzed, none of them had PGB.
About which cases are you talking? No reference is added. This is not clear.
13. 4) Most importantly, PGB is extremely rare, and the number of patients is small.
This is not a reason why GBC is less common now compared to the past. You should rephrase this to sample size is too low, as PGB is a rare disease, as such no reliable and significant statements can be made.
Reviewer 3 Report
I congratulate the authors for this manuscript which is well written and exhaustive
Author Response
Thank you very much for your great comment.